# Ethyl Acetate Fractions of *Salvia miltiorrhiza* Bunge (Danshen) Crude Extract Modulate Fibrotic Signals to Ameliorate Diabetic Kidney Injury

**DOI:** 10.3390/ijms25168986

**Published:** 2024-08-18

**Authors:** Yung-Chien Hsu, Ya-Hsueh Shih, Cheng Ho, Cheng-Chi Liu, Chia-Ching Liaw, Hui-Yi Lin, Chun-Liang Lin

**Affiliations:** 1Departments of Nephrology, Chang Gung Memorial Hospital, Chiayi 61363, Taiwan; libra@cgmh.org.tw (Y.-C.H.); rita1608@gmail.com (Y.-H.S.); gnoo0102@cgmh.org.tw (C.-C.L.); 2Kidney and Diabetic Complications Research Team (KDCRT), Chang Gung Memorial Hospital, Chiayi 61363, Taiwan; hc1238@cgmh.org.tw; 3School of Medicine, College of Medicine, Chang Gung University, Taoyuan 333423, Taiwan; 4Division of Endocrinology and Metabolism, Chang Gung Memorial Hospital, Chiayi 61363, Taiwan; 5National Research Institute of Chinese Medicine, Ministry of Health and Welfare, Taipei 112304, Taiwan; liawcc@nricm.edu.tw; 6School of Pharmacy, China Medical University, Taichung 406040, Taiwan; 7Center for Shockwave Medicine and Tissue Engineering, Kaohsiung Chang Gung Memorial Hospital, Chang Gung University College of Medicine, Kaohsiung 83301, Taiwan; 8Kidney Research Center, Chang Gung Memorial Hospital, Taipei 10507, Taiwan; 9School of Traditional Chinese Medicine, College of Medicine, Chang Gung University, Taoyuan 33302, Taiwan

**Keywords:** *Salvia miltiorrhiza*, ethyl acetate layer, diabetic nephropathy, mesangial cells, myofibroblast activation, renal fibrosis

## Abstract

Diabetic nephropathy, a leading cause of end-stage renal disease, accounts for significant morbidity and mortality. It is characterized by microinflammation in the glomeruli and myofibroblast activation in the tubulointerstitium. *Salvia miltiorrhiza* Bunge, a traditional Chinese medicine, is shown to possess anti-inflammatory and anti-fibrotic properties, implying its renal-protective potential. This study investigates which type of component can reduce the damage caused by diabetic nephropathy in a single setting. The ethyl acetate (EtOAc) layer was demonstrated to provoke peroxisome proliferator-activated receptor (PPAR)-α and PPAR-γ activities in renal mesangial cells by dual luciferase reporter assay. In a high glucose (HG)-cultured mesangial cell model, the EtOAc layer substantially inhibited HG-induced elevations of interleukin-1β, transforming growth factor-β1 (TGF-β1), and fibronectin, whereas down-regulated PPAR-γ was restored. In addition, among the extracts of *S. miltiorrhiza*, the EtOAc layer effectively mitigated TGF-β1-stimulated myofibroblast activation. The EtOAc layer also showed a potent ability to attenuate renal hypertrophy, proteinuria, and fibrotic severity by repressing diabetes-induced proinflammatory factor, extracellular matrix accumulation, and PPAR-γ reduction in the STZ-induced diabetes mouse model. Our findings, both in vitro and in vivo, indicate the potential of the EtOAc layer from *S. miltiorrhiza* for future drug development targeting diabetic nephropathy.

## 1. Introduction

Diabetic nephropathy (DN), the most devastating complication of chronic diabetes, often leads to end-stage renal disease and causes significant morbidity and mortality [1]. The incidence of dialysis-dependent diabetics has more than doubled over the past decades worldwide and continues to rise, although many efforts have been made to control the progression of DN [2]. Currently, there are still no effective treatments in preventing or attenuating diabetes-induced renal damage. Discovery of drugs targeting inflammatory and fibrotic signaling may be a promising therapeutic strategy.

Diabetic nephropathy is characterized by glomerular microarchitecture remodeling, myofibroblast recruitment, and subsequent tubulointerstitial fibrosis [3]. Many studies have provided evidence that various inflammatory molecules [4] and profibrotic factors [5,6] are involved in the progression of DN. Moreover, transforming growth factor-β1 (TGF-β1) has been suggested as a pivotal mediator to promote the accumulation of an extracellular matrix (ECM) in glomerular mesangial areas and myofibroblast transdifferentiation in DN [6]. Conversely, peroxisome proliferator-activated receptors (PPARs), particularly PPAR-γ and PPAR-α, have also been suggested to play crucial roles in renal physiology and pathophysiology [7,8]. Suppression of PPAR-γ and PPAR-α in diabetes has been linked to DN progression [9,10]. Despite the critical role of TGF-β1 in triggering hypertrophic and fibrogenic effects in diabetic nephropathy, there is still controversy about suppressing TGF-β1 as a potential treatment for DN. The dual effects of TGF-β1 in tissues make it uncertain whether its inhibition would be of benefit or be a detriment to DN [6]. Currently, PPAR-γ and PPAR-α agonists effectively delay and prevent the progression of various renal diseases, particularly diabetic nephropathy [11,12].

*Salvia miltiorrhiza* Bunge, also known as red sage or Danshen, has been used in traditional Chinese medicine for approximately 2000 years. In clinics, *S. miltiorrhiza* is often prescribed to treat cardiovascular, cerebral, and thrombotic diseases [13]. It is also reported to possess a wide range of biological effects, including antithrombotic, antioxidative, anti-inflammatory [13,14], and anti-fibrotic [15,16] activities, implying its renal-protective potential. In the past few years, researchers have focused on unraveling the potential benefits and exploring the active compounds of *S. miltiorrhiza* in treating kidney diseases. For instance, a nationwide retrospective cohort study provided evidence of the advantages of Chinese herbal medicines containing *S. miltiorrhiza* in improving survival rates among patients with chronic kidney disease [17]. Additionally, Salvianolic acid B, a main composition of *S. miltiorrhiza*, was reported to attenuate TGF-β1 mediated myofibroblastic induction in renal proximal tubular cells [18,19]. Pre-treatment with *S. miltiorrhiza* ethanol extract was also shown to prevent acute renal failure by reducing inflammatory cytokines and oxidative stress in rats with renal ischemia-reperfusion injuries [20]. Furthermore, intraperitoneal injection of Salvianolic acid A, a water-soluble constituent of *S. miltiorrhiza*, was found to ameliorate tubulointerstitial fibrosis in rats subjected to 5/6 nephrectomy [21]. On the other hand, recent studies have also reported the biological and therapeutic effects of *S. miltiorrhiza* on diabetes-associated complications [22,23]. Yin et al. showed that *S. miltiorrhiza* injection preserves renal function and reduces proteinuria in streptozotocin (STZ)-induced diabetic rats through anti-oxidative mechanisms [24]. Notably, emerging evidence suggests that *S. miltiorrhiza* may potentially treat DN. Tanshinone IIA, another compound from *S. miltiorrhiza*, was shown to inhibit epithelial-to-myofibroblast transdifferentiation in high glucose (HG)-treated renal proximal tubular cells [25]. Moreover, *S. miltiorrhiza* has been demonstrated to suppress inflammatory mediators and subsequent tubulointerstitial fibrosis in diabetic kidneys [26]. However, further investigations are needed to determine the specific contribution of the components from *S. miltiorrhiza* in impeding the progression of DN.

This study aimed to identify the potential practical components of alleviating DN. First, we extracted *S. miltiorrhiza* with 50% ethanol using refluxing and partitioning with four solvents of different polarities. Using two in vitro cell models, high glucose (HG)-treated renal mesangial cells and TGF-β1-stimulated renal fibroblasts, we found that the EtOAc layer of *S. miltiorrhiza* significantly suppressed proinflammatory and profibrotic signals while also restoring downregulated PPAR-γ, which is critical in the development of diabetic complications including DN. We also evaluated whether the EtOAc layer of *S. miltiorrhiza* could attenuate diabetes-induced kidney injuries using streptozotocin-induced diabetic mice. The results of the in vivo analysis validate the renoprotective effects of the *S. miltiorrhiza* EtOAc layer. We also evaluated the in vivo efficacy of the EtOAc layer in diabetic mice. Notably, the in vivo analysis results validate the renoprotective effects of the *S. miltiorrhiza* EtOAc layer.

## 2. Results

### 2.1. Effects of S. miltiorrhiza Extracts on PPAR-α and PPAR-γ Transcriptional Activities in Renal Mesangial Cells Using Dual-Luciferase Reporter Assay

Several pieces of evidence suggest that PPARs play a critical role, especially the -α and -γ isoforms, against oxidative stress [27,28,29], inflammatory responses [10,28,30], and fibrotic injuries [29,30] in diabetic nephropathy. To identify the components of *S. miltiorrhiza* potentially modulating PPAR activity, we partition the crude extract into four main fractions (as illustrated in Figure 1A). A dual luciferase reporter assay was used to evaluate extracts stimulating PPAR-α or PPAR-γ transcriptional activity [31]. Renal mesangial cells were transiently transfected with two luciferase constructs, PPRE X3-TK-luc and PPRE-pNL1.3, and then treated with *S. miltiorrhiza* extracts or a specific PPAR-α agonist, WY14643, and potent PPAR-γ agonist, rosiglitazone, as positive controls for their respective receptors, for 24 h. The reporter assay showed that PPAR-α transcription was significantly elevated in the treatment of EtOAc and in the n-hexane layer of *S. miltiorrhiza*, while minimal or no effects were observed in n-BuOH and the water layer (Figure 1B). The reporter assay showed a significant increase in PPAR-α transcription with EtOAc or the n-hexane layer of *S. miltiorrhiza*, while minimal to no effect was observed with n-BuOH and the water layer. On the other hand, among all *S. miltiorrhiza* extracts, only the EtOAc layer significantly induced the transcriptional activity of PPAR-γ (Figure 1C). These results suggest that the EtOAc layer of *S. miltiorrhiza* exhibited a strong activity in stimulating PPAR-α and PPAR-γ transcriptional activity in renal mesangial cells.

### 2.2. Effects of S. miltiorrhiza EtOAc Extract on Renal Inflammation/Fibrosis-Related Genes in HG-Stressed Mesangial Cells

The above results show the potential of the EtOAc extract of *S. miltiorrhiza* in activating PPAR-α and PPAR-γ transcriptional activity, which is crucial for protecting renal function and mitigating diabetic glomerular injury [32]. To investigate the effect of extracts of *S. miltiorrhiza* on diabetic glomerular injury, renal mesangial cells were treated with high glucose (HG, 30 mM D-glucose) to mimic high glucose-stressed glomerular injury in vitro. The *S. miltiorrhiza* extracts were co-treated with HG for 48 h, and the critical genes involved in glomerular injury response, including PPAR-γ, interleukin-1β (IL-1β), transforming growth factor-β1 (TGF-β1), and fibronectin, were examined. Consistent with previous reports [33,34], HG treatment increased TGF-β1, IL-1β, and fibronectin expression in mesangial cells compared to the mannitol control group. In contrast, PPAR-γ mRNA expression was down-regulated (Figure 2). The EtOAc extract of *S. miltiorrhiza* significantly inhibited HG-induced changes in these proinflammatory and profibrotic factors and restored PPAR-γ expression in renal mesangial cells. Notably, the EtOAc extract of *S. miltiorrhiza* showed a strong upregulation of PPAR gene expression compared to even the control group. These results indicate that the EtOAc extract of *S. miltiorrhiza* exhibited a potent activity in attenuating HG-induced glomerular injury.

### 2.3. Inhibitory Effects of S. miltiorrhiza Extracts on TGF-β1-Induced Myofibroblast Activation

TGF-β1 signaling pathway-triggered myofibroblast activation plays a critical role in renal fibrosis and the progression of diabetic nephropathy [6]. Hence, we investigated whether extracts of *S. miltiorrhiza* could also exhibit activity against TGF-β1-induced myofibroblast activation. In an in vitro renal fibrosis model, rat renal fibroblasts (NRK-49F cells) were treated with TGF-β1 (0, 1, 2, and 5 ng/mL) for 48 h, and α-smooth muscle actin (α-SMA), a key indicator of myofibroblast activation, was subsequently assessed [35]. Immunoblotting analysis revealed a significant increase in α-SMA levels at 5 ng/mL of TGF-β1 compared to other concentrations (Figure 3A), with maximal induction observed after 48 h of treatment with the same concentration (Figure 3B). The EtOAc and n-BuOH extracts of *S. miltiorrhiza* dose-dependently attenuated TGF-β1-induced upregulation of α-SMA proteins compared to the control group, with minimal effects observed for the n-hexane and water extracts (Figure 3C). Additionally, the n-BuOH extract exhibited weaker suppression of myofibroblast activation relative to the EtOAc fraction. These results suggest that the EtOAc extract of *S. miltiorrhiza* also exhibits potent activity in inhibiting TGF-β1-induced myofibroblast activation.

### 2.4. In Vivo Effects of EtOAc Extract of S. miltiorrhiza on Renal Gene Expression in STZ-Induced Diabetic Mice

Next, we investigated the in vivo activity of the EtOAc extract of *S. miltiorrhiza* using STZ-induced diabetic mouse models. After administering the extract to STZ-induced mice for 12 weeks, renal tissues were examined for PPAR-γ, IL-1β, and TGF-β1 gene expression. Immunohistochemical analysis showed upregulated IL-1β and TGF-β1 levels in diabetic mice, accompanied by reduced PPAR-γ expression. These results suggest that the EtOAc extract of *S. miltiorrhiza* significantly attenuated the diabetes-induced alterations of PPAR-γ, IL-1β, and TGF-β1 (Figure 4).

### 2.5. Renoprotective Effects of S. miltiorrhiza EtOAc Extract STZ-Induced Diabetic Mice

To gain further insight into the effects of the EtOAc extract on *S. miltiorrhiza*, we assessed it effects on kidney dysfunction and renal fibrosis in diabetic mice. Glycated hemoglobin (HbA1c) analysis revealed that treatment with *S. miltiorrhiza* EtOAc extract had minimal effect on glycemic control, whereas diabetic mice exhibited significantly higher HbA1c levels compared to normal controls (Figure 5A). On the other hand, treatment with *S. miltiorrhiza* EtOAc extract significantly inhibited the increase in relative kidney size, defined as the percentage of kidney weight to total weight, and daily urinary protein excretion in diabetic mice (Figure 5B,C). Moreover, immunohistochemical staining results of fibronectin, an extracellular matrix marker for evaluating fibrotic response, revealed that the EtOAc extract of *S. miltiorrhiza* significantly decreased diabetes-induced intense elevation of fibronectin (Figure 5D). Additionally, Masson’s trichrome staining showed that the *S. miltiorrhiza* EtOAc extract significantly attenuated collagen deposition in STZ-treated mice by 46% (Figure 5E). These results showed the significant renal protective activity of the *S. miltiorrhiza* EtOAc extract in STZ-induced mouse models.

### 2.6. HPLC Fingerprint Profile of the EtOAc Extract and the Isolation Components

To identify the components of the active EtOAc extract, the HPLC fingerprint profile (Figure 6) was established by RP-HPLC equipped with a PAD detector. Thirteen compounds, including caffeic acid, salvianolic acids A, B, E, F, G, and L, rosmarinic acid, lithospermic acid, 9′-methyl salvanolates B and H, yunnaneic acid, and salviaflaside, were isolated from the EtOAc extract of *S. miltiorrhiza*. These chemical structures were determined by spectroscopic data, including NMR and HRMS spectral data and compared with the reported data. Although the HPLC profile does not provide the precise percentage of each component, it offers insights into the potential effective component in the layer and indicates the direction for future research to identify the true effective component in the EtOAc extract.

## 3. Discussion

Diabetic nephropathy (DN) stands as a dire complication of chronic diabetes, often culminating in end-stage renal disease and significant morbidity and mortality [1]. Despite this, effective treatments to prevent or mitigate diabetes-induced renal damage remain elusive [36,37,38]. Recent studies have increasingly explored the biological and therapeutic effects of *S. miltiorrhiza* on diabetes-associated complications, including nephropathy [17,22,23]. However, compared to other diabetic complications, there is limited scientific evidence regarding the effects of *S. miltiorrhiza* on DN to date. In this study, we investigated the potential of *S. miltiorrhiza* extracts in treating diabetic nephropathy. Firstly, we purchased the *Salvia miltiorrhiza* from Chuang Song Zong Pharmaceutical Co., Ltd., Kaohsiung, Taiwan, a company recognized as a producer of Chinese medicine with cGMP certification. The extract of *S. miltiorrhiza* includes water-soluble and fat-soluble components. We utilized 50% alcohol to extract *S. miltiorrhiza* because the fat-soluble component would be significantly reduced if extracted with water. Furthermore, in our study, we chose to use crude extracts instead of serum-containing drugs to treat the cell line. This decision was based on the fact that crude extracts cannot be fully dissolved in serum due to their complex components. Incomplete dissolution may result in uneven concentrations, which could potentially interfere with the results of the *Salvia miltiorrhiza* treatment. Our results provide compelling experimental evidence that the EtOAc extract of *S. miltiorrhiza* exhibits potent activity in alleviating diabetic nephropathy. (1) The EtOAc extract significantly stimulated both PPAR-α and PPAR-γ transcriptional activities (Figure 1B,C), suggesting potential in mitigating inflammation, renal fibrotic injuries, and diabetic nephropathy, often associated with the suppression of PPAR-α or PPAR-γ [7,9]. (2) Notably, the EtOAc extract significantly inhibited high glucose-induced gene alterations associated with renal inflammation and fibrosis in renal mesangial cells (Figure 2). (3) The EtOAc extract could significantly suppress TGF-β1-induced myofibroblast activation (Figure 3), further supporting its potential as a therapeutic agent against renal fibrosis progression. (4) In vivo studies using STZ-induced diabetic mouse models elucidated the renal protective effects of the EtOAc extract, as evidenced by its ability to mitigate kidney dysfunction, reduce urinary protein excretion, and attenuate fibrotic responses (Figure 4 and Figure 5).

PPARs have been implicated in the pathogenesis of DN [9,10,28,29,30]. Treatment with PPAR-α agonist or activation of PPAR-αwere is demonstrated to attenuate DN in diabetic animals through anti-inflammatory and anti-fibrotic effects [39]. PPAR-α deficiency accelerates renal fibrosis and renal function decline in diabetic mice [40]. Conversely, PPAR-γ agonists, thiazolidinediones, have been extensively used in clinical treatments of type 2 diabetes for decades [10]. Our previous study demonstrated that restoration of PPAR-γ function protects kidneys against the hyperglycemic promotion of fibrogenic transcription factors and inflammation regulators [30]. In this study, we found that the EtOAc extract of *S. miltiorrhiza* significantly induced PPAR-α and PPAR-γ transcriptional activities in cultured renal mesangial cells (Figure 1B,C). Moreover, treatment with the EtOAc extract of *S. miltiorrhiza* could reverse down-regulated PPAR-γ expression in vitro and in vivo (Figure 2A and Figure 4). It has been reported that administration of *S. miltiorrhiza* could prevent PPAR-α suppression in a mouse model of alcoholic liver disease [41]. It has been reported before that Salvianolic acid B, a major polyphenolic compound of *S. miltiorrhiza*, significantly increased PPAR-α protein expression in the liver, improving glucose tolerance and dyslipidemia [42]. In contrast, Salvianolic acid B has been reported to improve steroid-induced osteonecrosis via inhibiting PPAR-γ2 in rats [43]. It has been reported that administration of *S. miltiorrhiza* could prevent PPAR-α suppression in a mouse model of alcoholic liver disease [41]. Salvianolic acid B, a major polyphenolic compound of *S. miltiorrhiza*, has been reported to significantly increase PPAR-α protein expression in the liver, improving glucose tolerance and dyslipidemia [42]. However, in rats, Salvianolic acid B has been reported to improve steroid-induced osteonecrosis by inhibiting PPAR-γ2 [43]. Despite significant effort being devoted by other researchers, we are the first group to reveal the effects of *S. miltiorrhiza* on modulating PPAR-α or PPAR-γ expressions in HG-cultured renal cells or diabetic mice from the gene through the protein levels.

Diabetic kidney injury is also associated with the recruitment of inflammatory cells into the glomerular mesangium and tubulointerstitium [3]. The influx of inflammatory cells and their related cytokines are responsible for the ongoing fibrogenic processes of DN/Diabetic kidney. IL-1β, a critical proinflammatory cytokine, enhances the chemotaxis of inflammatory cells, increases vascular endothelial permeability, and may promote the onset and progression of diabetic kidney disease [44]. Previous studies have demonstrated the anti-inflammatory properties of *S. miltiorrhiza* [21,26,45]. The present study showed that EtOAc extract of *S. miltiorrhiza* significantly suppressed HG-induced IL-1β gene expression in vitro (Figure 2B) and in vivo (Figure 4). These results suggest that the EtOAc extract of *S. miltiorrhiza* may attenuate diabetic kidney injury by suppressing critical inflammatory cytokines such as IL-1β.

The progression of ECM accumulation induced by high glucose stress is also implicated in the pathogenesis of diabetic nephropathy [46]. TGF-β1 signaling is a well-established key mediator that promotes ECM accumulation in the mesangium and drives myofibroblast activation during the development of renal fibrosis in the diabetic milieu [6,47,48]. It has been reported that *S. miltiorrhiza* and its derivatives showed inhibitory activity on the TGF-β1/Smads pathway to suppress myocardial injury, hepatocarcinogenesis, and pulmonary fibrosis [16,33,34]. The present study found that the EtOAc extract of *S. miltiorrhiza* significantly suppressed TGF-β1-induced myofibroblast activation (Figure 3A,B). In addition, in the HG-stimulated renal mesangial cell model, the EtOAc extract of *S. miltiorrhiza* significantly inhibited fibronectin’s gene expression, an essential ECM composition (Figure 2C,D). In STZ-induced diabetic mouse models, the EtOAc extract of *S. miltiorrhiza* also significantly reduced TGF-β1 gene expression (Figure 4B), fibronectin protein expression (Figure 5D), and collage deposition (Figure 5E). These observations suggest that the renal protective activity of the EtOAc extract of *S. miltiorrhiza* may involve the suppression of TGF-β1 signaling-induced ECM accumulation, at least in part.

Several animal studies have demonstrated the hypoglycemic effects of *S. miltiorrhiza* and its ingredients [9,35,39]. However, some studies did not reveal the anti-diabetic effect of *S. miltiorrhiza* and its derivatives. For instance, oral administration of a water extract of *S. miltiorrhiza* (500 mg/kg/day) into STZ-treated rats failed to improve glycemic conditions [40]. Our results also indicate that the EtOAc extract of *S. miltiorrhiza* did not improve glycated hemoglobin (HbA1c) levels compared to the glycemic control (Figure 5A). Similarly, Xu et al. showed that intraperitoneal injection of *S. miltiorrhiza* at a dosage of 0.5 or 1 mL/kg/day for 6 weeks did not change blood glucose levels in STZ-induced diabetic rats [26]. The discrepancy in the efficacy of *S. miltiorrhiza* in glycemic regulation may be attributed to the differences in animal models, administration routes, drug compositions, and treatment dosages. These observations indicated that *S. miltiorrhiza* or its extracts ameliorate DN by regulating additional pathogenic pathways beyond hyperglycemia.

In the present study, we extracted *S. miltiorrhiza* using several polar solvents, including n-hexane, EtOAc, *n*-butanol, and water. Our results revealed that the EtOAc extract exhibited the highest potency in preventing diabetic renal fibrosis in vitro (Figure 2C,D and Figure 3C). Furthermore, we validated the therapeutic effect of EtOAc of *S. miltiorrhiza* on DN in an STZ-induced mouse model. Our results demonstrated that the *S. miltiorrhiza* EtOAc extract exerts antifibrotic activities by repressing hyperglycemia-induced proinflammatory factor, ECM accumulation, and PPAR-γ reduction (Figure 4 and Figure 5). To investigate the antifibrotic properties of the *S. miltiorrhiza* EtOAc extract and its potential to restore PPAR-γ, we subjected the EtOAc layers to a comprehensive component analysis. Thirteen distinct components were isolated, and further research is essential to determine which component is responsible for the antifibrotic effect (Figure 6). Collectively, these results suggest that the EtOAc extract of *S. miltiorrhiza* has a potential role in alleviating the progression of diabetic nephropathy through down-regulation of TGF-β1 signaling and suppression of ECM deposition (Figure 7). Our results indicate that the *S. miltiorrhiza* chromatogram found that the EtOAc extract contains multiple polyphenolic compounds that may have the potential for drug development against diabetic renal fibrosis in the future.

## 4. Materials and Methods

### 4.1. Cell Cultures

#### 4.1.1. In Vitro Model of High Glucose Stress

Renal mesangial cells (ATCC #CRL-2573TM), an immortalized rat mesangial cell line, were obtained from the American Type Culture Collection (Manassas, VA, USA). The cells were maintained in Dulbecco’s modified Eagle’s medium (DMEM) containing 10% fetal bovine serum (Gibco, Carlsbad, CA, USA), 100 U/mL penicillin, and 100 mg/mL streptomycin at 37 °C in a 5% CO_2_ humidified atmosphere. In our study, to prevent potential senescence and possible genetic and morphological changes compared to the parental cells, we set the passage number of our cells to 5 (P17–P21). To mimic the hyperglycemic milieu, renal mesangial cells were cultured with high concentrations of *D*-glucose (30 mM) for 48 h, and *D*-mannitol served as an osmotic control.

#### 4.1.2. In Vitro Model of Myofibroblast Activation

The rat kidney fibroblast cell line NRK-49F (ATCC #CRL-1570TM) was purchased from the American Type Culture Collection (Manassas, VA, USA). The cells were cultured in DMEM supplemented with 2 mM L-glutamine, 1% non-essential amino acids, and 5% fetal bovine serum (Gibco, Carlsbad, CA, USA) in a 5% CO_2_, 37 °C humidified incubator. To establish an in vitro model of renal fibrosis, subconfluent NRK-49F cells were made quiescent in basal medium with 0.5% fetal bovine serum for 24 h and then stimulated with 5 ng/mL TGF-β1 (PeproTech, Rocky Hill, NJ, USA) for another 48 h.

### 4.2. Preparation of the S. miltiorrhiza (Danshen) Extract

Dried roots of *Salvia miltiorrhiza* were purchased from Chuang Song-Zong Pharma- ceutical Co., Kaohsiung, Taiwan. The preparation and extraction procedures are illustrated in Figure 1A. Slices of *S. miltiorrhiza* (200 g) were pulverized and refluxed with 50% ethanol. After evaporation of the solvent, the crude extract was further partitioned with n-hexane (n-hexane), ethyl acetate (EtOAc), and *n*-butanol (n-BuOH). All of the liquid extracts were concentrated using a rotary evaporator (Yamato Scientific Co., Tokyo, Japan) until they were dry. Dried extracts were then dissolved in cell culture-grade DMSO (Sigma Aldrich Inc., St. Louis, MO, USA) to prepare a 20 mg/mL stock solution and stored at 4 °C. The *S. miltiorrhiza* extract was diluted with culture media to achieve the indicated final concentration in each experiment. DMSO, used as the vehicle control, was added in an equal amount and was less than 2.5‰ (*v*/*v*).

### 4.3. Main Compounds Isolation and HPLC Profile of the EtOAc Extract

The HPLC fingerprint profile of the EtOAc extract was established by Shimadza Nexera-i LC-2040C 3D UHPLC system equipped with a COSMOSIL 5C18-ARII column (250 × 4.6 mm, 5 μm) kept at 25 °C and yielded 13 major components. The mobile phase flow rate and the injection volume were 1.0 mL/min and 10 μL (10 mg/mL), respectively. The mobile phase consisted of H_2_O (A) and acetonitrile (B), both with 0.3% phosphoric acid, using a gradient program of separation conditions as follow: 0.01–45.00 min, 5–50% B, 45.00–50 min, 50–65% B, 50–75 min, 65–100% B, and 75–85 min, 100% B. The UV wavelength was set at 280 nm.

### 4.4. Measurement of PPAR-α and PPAR-γ Activities by Reporter Assay

For analysis of PPAR-α and PPAR-γ response element (PPRE) promoter activity, luciferase constructs, PPRE X3-TK-luc (Addgene plasmid #1015; Cambridge, MA, USA) and PPRE-pNL1.3 (Addgene plasmid #84394; Cambridge, MA, USA) were used, respectively. One day before transfection, 5 × 10^4^ renal mesangial cells diluted in 1 mL of DMEM media were seeded in each well of a 24-well plate. When cells reached ~90% confluency, the reporter plasmids PPRE X3-TK-Luc or PPRE-pNL1.3 were co-transfected with the Renilla luciferase expression vectors pRL-TK or pNL1.3 (Promega, Madison, WI, USA) using Lipofectamine 2000 Transfection Reagent (Invitrogen, Carlsbad, CA, USA). The transiently transfected renal mesangial cells were then treated with 20 μM of WY14643 (a specific PPAR-α agonist; Sigma Aldrich Inc., St. Louis, MO, USA), 1 μM of rosiglitazone (BRL-49653) (a potent PPAR-γ agonist; Sigma Aldrich Inc., St. Louis, MO, USA), or 20 μg/mL of each *S. miltiorrhiza* crude extract for 24 h. After experiments, cell lysates were analyzed using the dual luciferase reporter assay system (Promega, Madison, WI, USA). The PPRE luciferase activity served as an indicator of PPAR-α or PPAR-γ activation and was normalized to Renilla luciferase expression.

### 4.5. RNA Extraction and Quantitative Reverse Transcription-PCR (qRT-PCR)

The expression of PPAR-γ, IL-1β, TGF-β1, and fibronectin genes was detected by qRT-PCR. Total RNAs from cultured cells were extracted using QIAzol lysis reagent (Qiagen, Valencia, CA, USA) according to the manufacturer’s instructions. One microgram of total RNA was reverse-transcribed into first-strand cDNA with the ReverAidTM M-MuLV reverse transcriptase (Fermentas, Glen Burnie, MD, USA). Twenty-five microliters of PCR mixtures containing cDNA templates (equivalent to 20 ng of total RNA), 2.5 μM of specific forward and reverse primers, and 2× iQTM SYBR Green Supermix (Bio-Rad Laboratories, Hercules, CA, USA) were amplified using the iCycler iQ Real-time PCR Detection System (Bio-Rad Laboratories, Hercules, CA, USA). The thermocycler setting for PCR amplification was the initial melting step at 95 °C for 5 min, followed by 35 cycles of denaturation at 94 °C for 15 s, annealing at 52 °C for 20 s, and extension at 72 °C for 30 s. The annealing temperature was set according to the oligonucleotide data sheet provided by the primer manufacturer. The reference temperatures are 51.8 °C and 53.8 °C for different sequences. The primer sequences used in this study are shown in Table 1. All qRT-PCR reactions were performed in duplicate for at least three independent experiments. The relative gene expression was calculated as previously described and according to the MIQE guidelines. The amplification efficiencies of qPCR ranged from 95% to 105% for all the tested primer sets [49]. Relative changes in gene expression were calculated as previously described [30].

### 4.6. Protein Extraction and Western Blot Analysis

Cell proteins were extracted from cultured renal cells according to our previously described protocols [50]. Equal amounts of protein lysates were separated by 8–12% SDS-polyacrylamide gel electrophoresis (SDS-PAGE). After gel electrophoresis, the separated proteins were transferred onto a polyvinylidene difluoride (PVDF) membrane (Bio-Rad Laboratories, Hercules, CA, USA) and were then probed with specific primary IgG antibodies against α-SMA at a 1:1000 dilution (Abcam, Cambridge, UK, ab7817), followed by goat anti-rabbit horseradish peroxidase-conjugated IgG (Santa Cruz Biotechnology, Dallas, TX, USA) as the secondary antibody, and visualized by chemiluminescence. Subsequently, these membranes were stripped and re-probed with IgG antibodies against β-actin at a 1:1000 dilution (Cell Signaling Technology, Beverly, MA, USA, #9470) with the same procedures to show equal loading.

### 4.7. Streptozotocin (STZ)-Induced Diabetic Animal Model and Treatment

Three-month-old male C57BL/6 mice (BioLasco Biotechnology Co., Taipei, Taiwan) were intraperitoneally injected with a single dose of 190 mg/kg STZ (Sigma Aldrich Inc., St. Louis, MO, USA) to induce diabetes. Each diabetic mouse was given 1–2 units/kg of intermediate-acting insulin once daily (Monotard; Novo Nordisk A/S, Bagsvaerd, Denmark) to equalize glycemic control as previously described [51]. One week after injection, mice with a fasting blood glucose of 200–300 mg/dL were considered as diabetic and selected for further studies. A total of 24 mice were randomly subdivided into the following experimental groups, including normal controls (n = 6), the diabetic group (n = 6), the vehicle-treated diabetic group (n = 6), and the *S. miltiorrhiza* EtOAc extract-treated diabetic group (n = 6). For in vivo administration of the *S. miltiorrhiza* EtOAc extract, mice were treated daily with a dose of 10 mg/kg intraperitoneal injection after successful induction of diabetes.

The reason we selected 10 mg/kg as our intraperitoneal injection dose is that after our dose dependent trial, 10 mg/kg dose was the minimal dose that yielded the maximum effect. After 12 weeks of treatment, mice were euthanized by intraperitoneal injection of pentobarbital sodium, and the kidneys were dissected for further studies. Animal experiments were performed at the Laboratory Animal Center, Department of Medical Research, Chang Gung Memorial Hospital at Chiayi. The Laboratory Animal Center is accredited by the Association for the Assessment and Accreditation of Laboratory Animal Care International (AAALAC) and has a full-time veterinarian. Animal cages were limited to 2 mice per cage, and mice were free to access food and water. All protocols for animal use and experiments were approved by the Institutional Animal Care and Use Committee of the Chang Gung Memorial Hospital and were performed according to the Animal Protection Law by the Council of Agriculture, Executive Yuan (R.O.C.) and the guidelines of the Nation Institutes of Health (Bethesda, MD, USA) for the care and use of laboratory animals.

### 4.8. Urine and Blood Biochemistry

Peripheral blood and urine samples were collected to evaluate renal function in mice. Serum levels of glycated hemoglobin A1c (HbA1c; Primus Diagnostics, Kansas City, MO, USA) and blood glucose (Dade Behring Inc., Newark, NJ, USA) were determined according to the manufacturer’s instructions. The daily urine excretion of each animal was collected using a metabolic cage system. Urinary protein (Dade Behring Inc., Newark, NJ, USA) and creatinine (Formosa Biomedical Technology Corp, Taipei, Taiwan) were measured using the respective assay kits. Urinary protein excretion was normalized to urinary creatinine levels. In addition, mice kidneys were removed at termination and weighed. The kidney-to-body weight ratio was calculated.

### 4.9. Histological and Immunohistochemical Examinations

After being adequately perfused, harvested kidneys were fixed in 4% PBS buffered formaldehyde, embedded in paraffin, and then sliced longitudinally into 4 μm thick sections. For immunohistochemical staining, renal specimens were incubated with primary antibodies against PPAR-γ, IL-1β, fibronectin (Abcam, Cambridge, UK), and TGF-β1 (Santa Cruz Biotechnology, Dallas, TX, USA) overnight at 4 °C. After washing with PBS solution, the slides were incubated with a secondary antibody for 60 min at 37 °C. Immunoreactivity in sections was then detected using a horseradish peroxidase-3′-,3′-diaminobenzidine kit (SuperPictureTM Kit; Invitrogen, Carlsbad, CA, USA) according to the manufacturer’s instructions. Furthermore, renal tissue sections were subjected to Masson’s trichrome stain kit (Sigma Aldrich Inc., St. Louis, MO, USA) to assess the degrees of renal fibrosis. The blue-stained area in the glomerulus or tubulointerstitium was selected as the positive area for collagen accumulation. For histomorphometric analysis, a digital light microscope (Carl Zeiss, Gottingen, Germany) was used to capture images. The Image-Pro^®^ Plus image analysis software (Media Cybernetics, Silver Spring, MD, USA) was used to analyze 10 randomly selected fields from each section (3 sections for each mouse) at 400× magnification for determining the positive area of immunohistochemical (brown stain) or trichrome (blue stain) stain.

### 4.10. Statistical Analysis

All values are expressed as mean ± standard deviation. The Mann–Whitney U test was used to evaluate differences between the sample of interest and its respective control. One-way analysis of variance (ANOVA) followed by the Bonferroni post-hoc test was performed to analyze the differences between the different treated groups. All statistical analyses were performed using SPSS Statistics for Windows, Version 19.0. (IBM Corp., Armonk, NY, USA). A *p* value less than 0.05 was considered statistically significant.

## Figures and Tables

**Figure 1 ijms-25-08986-f001:**
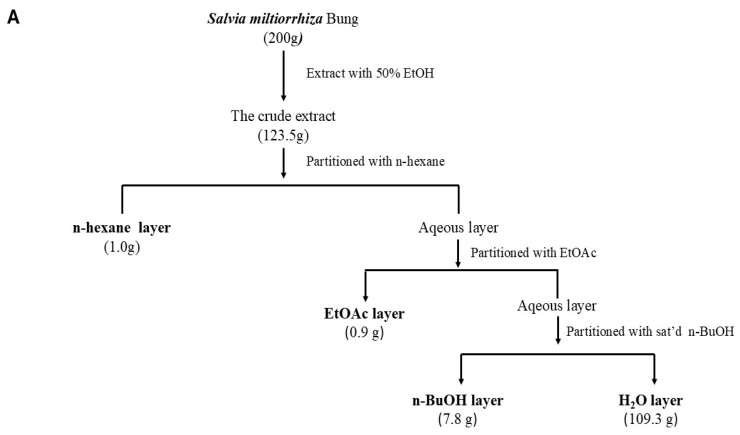
Ethyl Acetate extract of *S. miltiorrhiza* induces peroxisome proliferator-activated receptor (PPAR)-α and PPAR-γ transcriptional activities in renal mesangial cells. (**A**) Flow chart for preparation of n-hexane, ethyl acetate, *n*-butanol, and water extracts from *S. miltiorrhiza*. (**B**) Measurement of PPAR-α activity by luciferase reporter assay in transiently transfected renal mesangial cells that were treated with 20 μM WY14643 (a specific PPAR-α agonist) and different *S. miltiorrhiza* extract (20 μg/mL). (**C**) Measurement of PPAR-γ activity by luciferase reporter assay in transfected renal mesangial cells that were treated with 1 μM BRL-49653 (a specific PPAR-γ agonist) and each *S. miltiorrhiza* extract (μg/mL). The results are expressed as the means ± SD of six independent experiments. * *p* < 0.05 versus the normal control.

**Figure 2 ijms-25-08986-f002:**
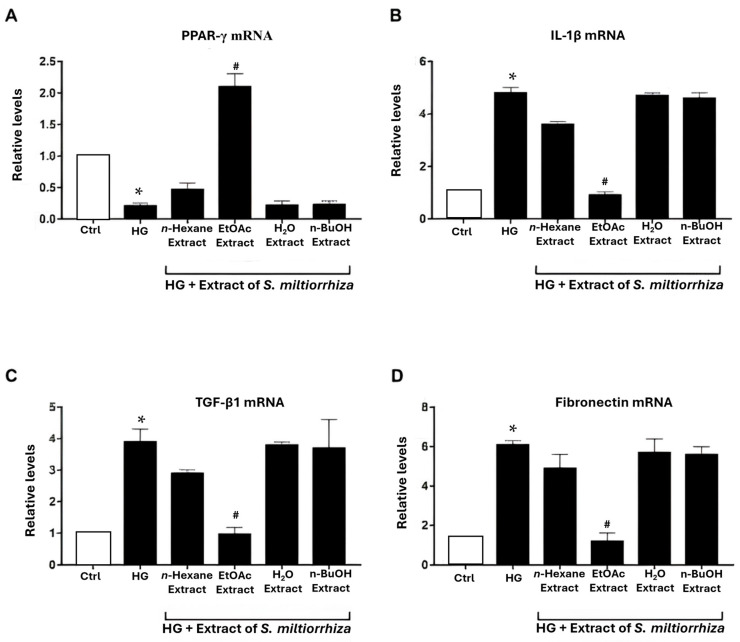
The effects of different extracts of *S. miltiorrhiza* on PPAR-γ, interleukin-1β (IL-1β), transforming growth factor-β1 (TGF-β1), and fibronectin gene expressions in high glucose (HG)-stimulated renal mesangial cells. Renal mesangial cells cultured with HG (30 nM D-glucose) were co-incubated with different *S. miltiorrhiza* extracts for 48 h. Renal mesangial cells treated with osmotic control (30 nM mannitol) act as the control group. The mRNA expression levels of (**A**) PPAR-γ, (**B**) IL-1β, (**C**) TGF-β1, and (**D**) fibronectin, normalized to that of β-actin, in these treated cells were determined by quantitative RT-PCR (n = 6). The symbol * indicates a significant difference compared to the osmotic controls (*p* < 0.05); the symbol # indicates a significant difference compared to the HG-cultured cells (*p* < 0.05).

**Figure 3 ijms-25-08986-f003:**
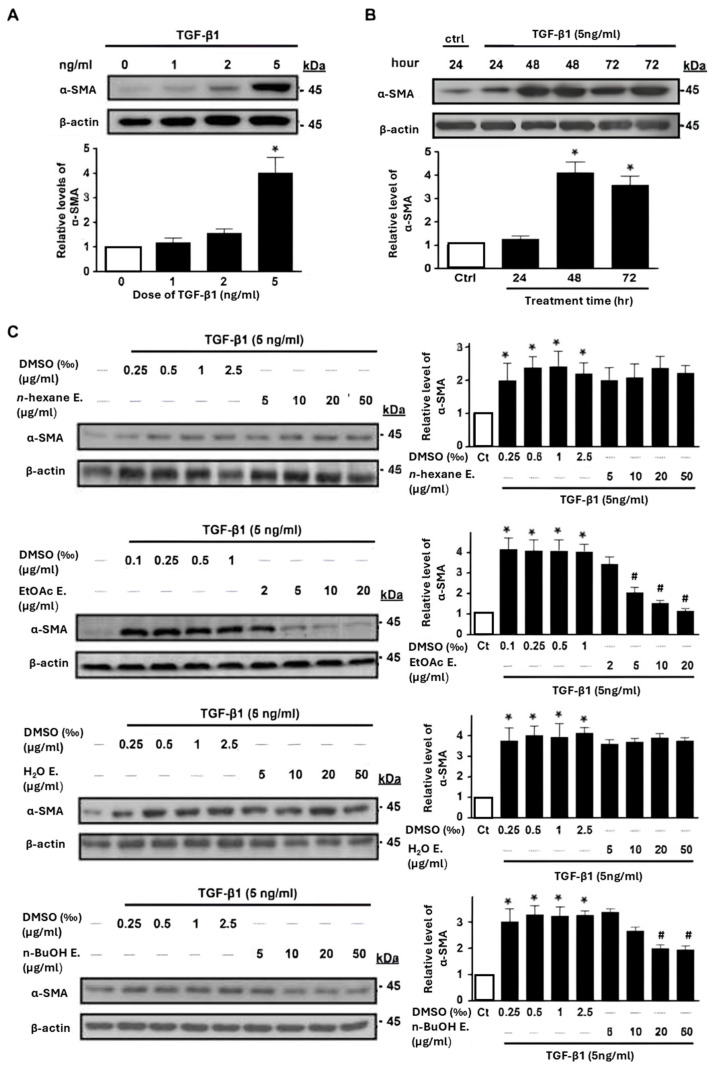
EtOAc extract of *S. miltiorrhiza* inhibits α-smooth muscle actin (α-SMA) up-regulation in TGF-β1-treated fibroblasts most effectively. (**A**) Effect of increasing TGF-β1 doses on the expression of α-SMA, a pivotal marker of myofibroblast activation, in NRK-49F cells. * *p* < 0.05 compared to normal controls. (**B**) Time-course effect of TGF-β1 (5 ng/mL) on the promotion of α-SMA activation in NRK-49F cells. * *p* < 0.05 versus normal controls for the indicated time points. (**C**) Western blot analysis of α-SMA expression in TGF-β1-stimulated NRK-49F cells co-cultured with different concentrations of *S. miltiorrhiza* extracts or the solvent control (dimethyl sulfoxide; DMSO) for 48 h. * *p* < 0.05 compared to the normal controls, # *p* < 0.05 versus DMSO-treated controls. Protein lysates from treated renal fibroblasts were analyzed for α-SMA expression by immunoblotting, followed by densitometric quantification. Data are presented as means ± SD (n = 3 for each experiment).

**Figure 4 ijms-25-08986-f004:**
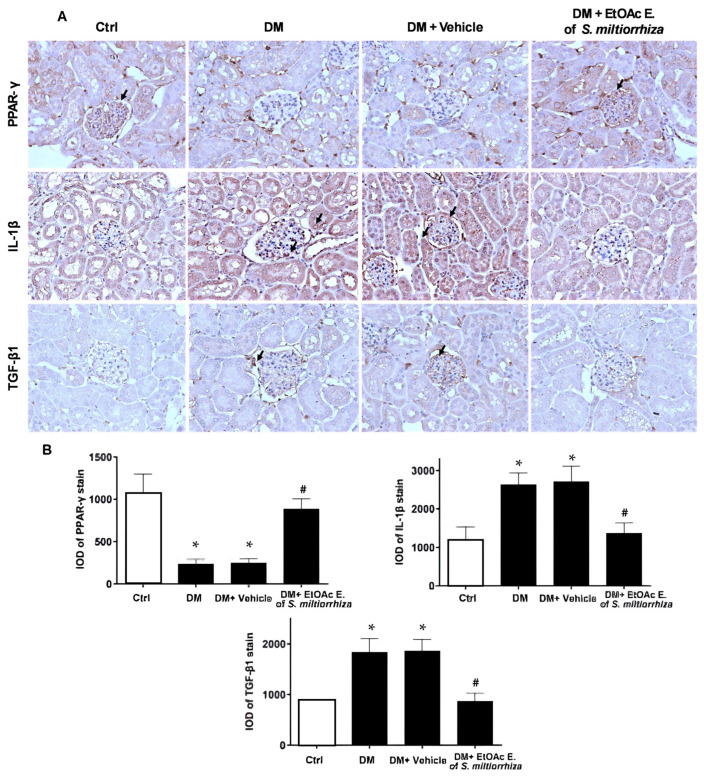
Treatment with *S. miltiorrhiza* EtOAc extract reduces expressions of proinflammatory and profibrotic signals and restores the PPAR-γ expression in streptozotocin (STZ)-induced diabetic mice. (**A**) Representative immunohistochemical (IHC) staining of PPAR-γ, IL-1β, and TGF-β1 in renal sections from normal control, diabetes (DM), vehicle-treated DM, and *S. miltiorrhiza* EtOAc extract-treated DM mice. (Arrows indicate areas of increased expression of PPAR-γ, IL-1β, and TGF-β1). Magnifications: ×400. (**B**) The integrated optical density (IOD) of PPAR-γ, IL-1β, and TGF-β1 IHC-stains were analyzed. Values are presented as means ± SD (n = 6). The symbol * indicates a significant difference compared to the normal control group (*p* < 0.05); the symbol # indicates a significant difference compared to the DM group (*p* < 0.05).

**Figure 5 ijms-25-08986-f005:**
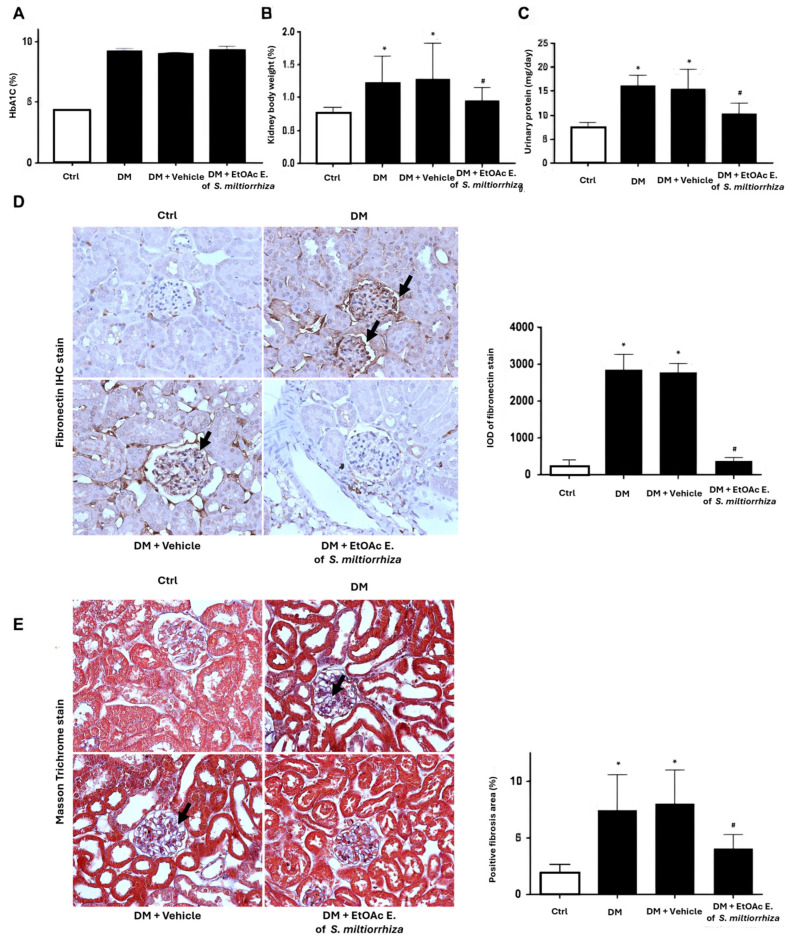
Treatment with the EtOAc extract from *S. miltiorrhiza* protects against diabetes-induced kidney injury. (**A**) Levels of glycated hemoglobin A1c (HbA1c), (**B**) kidney to total body weight ratio, and (**C**) daily excretion of urinary total protein in normal control, DM, vehicle-treated DM, and *S. miltiorrhiza* EtOAc extract-treated DM mice. (**D**) Representative IHC staining of fibronectin in renal sections from normal control, DM, and DM treated with vehicle or EtOAc extract of *S. miltiorrhiza*. (Arrow indicate areas of increased expression of fibronectin). The IOD of fibronectin-positive areas was analyzed, and data are presented as means ± SD. (**E**) The severity of fibrosis and extent of collagen deposition is evaluated by Masson’s trichrome staining (magnification × 400) (Arrow indicate areas of increased expression of fibronectin). The positive area of fibrosis is significantly diminished after treatment with *S. miltiorrhiza* EtOAc extract in DM mice. * *p* < 0.05 versus normal controls; # *p* versus STZ-induced DM mice (n = 6).

**Figure 6 ijms-25-08986-f006:**
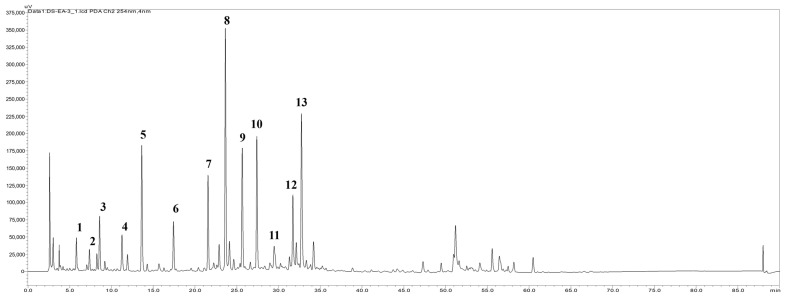
The HPLC fingerprint profile of EtOAc extract at 280 nm. caffeic acid (1), salvianolic acid F (2), yunnaneic acid (3), salviaflaside (4), salvianolic acid G (5), salvianolic acid E (6), rosmarinic acid (7), lithospermic acid (8), salvianolic acid B (9), salvianolic acid L (10), salvianolic acid A (11), 9′-methyl salvanolate B (12), and 9′-methyl salvanolate H (13).

**Figure 7 ijms-25-08986-f007:**
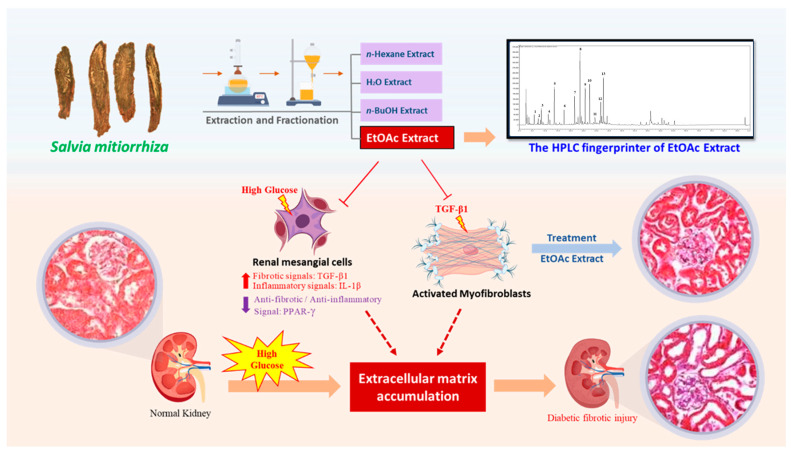
Schematic illustration of possible mechanisms of the *S. miltiorrhiza* EtOAc extract in the treatment of diabetic renal damage. The EtOAc extract in this study is shown to suppress fibrotic and inflammatory signals and restore decreased PPAR-γ expression in the diabetic milieu. Specimens were observed under ×400 magnification.

**Table 1 ijms-25-08986-t001:** Primer sequences used for quantitative RT-PCR.

Genes	Sense (Forward)	Antisense (Reverse)
PPAR-γ	5′-CCAAAGCCTGAGCCCAGA-3′	5′-GCACCACTCCCATGGCAT-3′
IL-1β	5′-CCAGGATGAGGACCCAAGCA-3′	5′-TCCCGACCATTGCTGTTTCC-3′
TGF-β1	5′-TGAGTGGCTGTCTTTTGACG-3′	5′-TGGGACTGATCCCATTGATT-3′
Fibronectin	5′-CAGCCCCTGATTGGAGTC-3′	5′-TGGGTGACACCTGAGTGAAC-3′
β-actin	5′-CGCCAACCGCGAGAAGAT-3′	5′-CGTCACCGGAGTCCATCA-3′

## Data Availability

The datasets used and/or analyzed during the current study are available from the corresponding authors on reasonable request.

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
