# Peer review of "Ethyl Acetate Fractions of Salvia miltiorrhiza Bunge (Danshen) Crude Extract Modulate Fibrotic Signals to Ameliorate Diabetic Kidney Injury"

_ijms, 2024, doi:10.3390/ijms25168986_

Round 1

Reviewer 1 Report

Comments and Suggestions for Authors

In vitro should be written italic font throughout the manuscript.

Please confirm if Figure 7 was created solely by the authors.

Please justify why water extraction was not used for Salvia miltiorrhiza .

DOI links should be provided for all cited references.

Page 15: Line 508: Ethical approval (Code) for the animal study needs to be mentioned. Any animal work needs ethical approval from the designated committee. Otherwise the manuscript cannot be published!

(https://www.ncbi.nlm.nih.gov/pmc/articles/PMC9710398/)

Figures 1 & 2 7 3 & 4 (Panel B): Please use white bar for the control group.

Figure 4: Please use arrows for different comparisons in panel A.

Figure 5: Please use arrows for different comparisons in panel E.

Figure 6: Why red font was used for the highlighted peaks.

What is the passage number used for the cultured cells.

Page 12: Delete 364: Line This section is not mandatory but can be added to the manuscript if the discussion is 364 unusually long or complex.

Comments on the Quality of English Language

Only minor English revision is required.

Author Response

Author’s reply to Reviewer 1

Comment 1: In vitro should be written in italic font throughout the manuscript.

Response 1: We are thankful for the reviewer's precious advice. We will correct the usage of "in vitro” in the draft to the proper format as per your advice.

Comment 2: Please confirm if Figure 7 was created solely by the authors.

Response 2: We appreciate the reviewer for the valuable feedback. Our authors and research team created this image, and there are no concerns regarding plagiarism or duplicating existing images. Therefore, we affirm that the image was exclusively created by the authors of this article.

Comment 3: Please justify why water extraction was not used for Salvia miltiorrhiza.

Response 3: We appreciate the reviewer's opinion. The components of Salvia miltiorrhiza include water-soluble and fat-soluble components. If it was extracted with water, the amount of fat-soluble components extracted will be greatly reduced. After consideration, we were thought that the active ingredients should be extracted as much as possible, so we used 50% alcohol to extract them. We add the explanation to this manuscript on lines 280-282.

Comment 4: DOI links should be provided for all cited references.

Response 4: We apologize for not providing DOI links. We will provide the DOI link in this manuscript for all cited references.

Comment 5: Page 15: Line 508: Ethical approval (Code) for the animal study needs to be mentioned. Any animal work needs ethical approval from the designated committee. Otherwise, the manuscript cannot be published!

Response 5: thanks for your previous remind. Our approval number for animal use in our study is 2019082301 and 2021112502. We will add the information to this manuscript on line 550-552.

(https://www.ncbi.nlm.nih.gov/pmc/articles/PMC9710398/)

Comment 6: Figures 1 & 2 7 3 & 4 (Panel B): Please use white bar for the control group.

Response 6: We appreciate the reviewer's opinion. We will adjust the image and use a white bar for the control group.

Comment 7: Figure 4: Please use arrows for different comparisons in panel A.

Response 7: We appreciate the reviewer's opinion. We will add arrows to indicate the different comparisons.

Comment 8: Figure 5: Please use arrows for different comparisons in panel E.

Response 8: We appreciate the reviewer's opinion. We will add arrows to indicate the different comparisons.

Comment 9: Figure 6: Why red font was used for the highlighted peaks.

Response 9: We sincerely apologize for not thoroughly reviewing the content, which resulted in this typographical error. We have corrected the red text back to black.

Comment 10: What is the passage number used for the cultured cells?

Response 10: Thank you for the reviewer's pivotal question. In our study, to prevent potential senescence and possible genetic and morphological changes compared to the parental cells, we set the passage number of our cells to 5 (P17-P21). We added information of the passage number on lines: 393-395

Comment 11: Page 12: Delete 364: Line This section is not mandatory but can be added to the manuscript if the discussion is 364 unusually long or complex.

Response 11: We sincerely apologize for not thoroughly reviewing the content, which resulted in the appearance of redundant phrases. We have corrected them accordingly.

Reviewer 2 Report

Comments and Suggestions for Authors

1. The image clarity is too low.

2. The author only conducted component analysis on the ethanol extract. Why didn't they analyze the components of other extraction methods to identify differential pharmacological ingredients?

3. The antibodies used should be clearly labeled with their catalog numbers, dilution factors, and other relevant information.

4. The lesions should be marked on the pathological images in accordance with standard practices.

5. Has the purchased Salvia miltiorrhiza been identified by a specialist in pharmacognosy?

6. The annealing temperature of 52 degrees Celsius for qPCR seems somewhat low. It's common practice to perform a temperature gradient to explore the optimal annealing temperature for the specific primers being used. This ensures efficient amplification and accurate results.

7. Does the presentation of qPCR data comply with MIQE standards?

8. There are various treatment methods when conducting in vitro experiments
with crude extracts of natural medicines. Direct application of crude extracts to cell lines may not mimic
the physiological processes of drugs in vivo. Why didn't the authors choose to use serum containing the drug
for treating cell lines?

Author Response

Author’s reply to Reviewer 2

Comment 1: The image clarity is too low.

Response 1: Thank you for your valuable feedback. We will improve the image clarity.

Comment 2: The author only conducted component analysis on the ethanol extract. Why didn't they analyze the components of other extraction methods to identify differential pharmacological ingredients?

Response 2: Thanks for your precious comments. Firstly, the reason we use the EtOAc layers for component analysis is that the EtOAc layers have been proven to efficiently inhibit HG-induced elevations of interleukin-1β, transforming growth factor-β1 (TGF-β1), and fibronectin, and restore PPAR-γ. They also show attenuation of renal hypertrophy, proteinuria, and fibrotic severity in the STZ-induced diabetes mouse model compared to other layers in our study. Therefore, we sent EtOAc layers for component analysis to identify the main component of this layer. Through further research on these components, we can more accurately determine which components  can inhibit fibrosis in EtOAc layers. We incorporate this response into our manuscript on lines 370-374

Comment 3: The antibodies used should be clearly labeled with their catalog numbers, dilution factors, and other relevant information.

Response 3: We apologize for the incomplete information regarding the antibodies used in our study. The relevant information including catalog numbers, and dilution factors will be added to the manuscript on lines: 471-476

Comment 4: The lesions should be marked on the pathological images in accordance with standard practices.

Response 4: Thanks for your previous feedback. We will label the lesions on the pathological image with standard practices.

Comment 5: Has the purchased Salvia miltiorrhiza been identified by a specialist in pharmacognosy?

Response 5: We apologize for not providing the important information, and we thank the reviewer for the reminder. We purchased the Salvia miltiorrhiza from Chuang Song Zong Pharmaceutical Co., Ltd., a company recognized as a producer of Chinese medicine with cGMP certification. As a result, we trust that this company can properly identify the medicine. The information was added to our manuscript on lines 278-280

Comment 6: The annealing temperature of 52 degrees Celsius for qPCR seems somewhat low. It's common practice to perform a temperature gradient to explore the optimal annealing temperature for the specific primers being used. This ensures efficient amplification and accurate results.

Response6: We are thankful for the reviewer's pivotal viewpoint. We checked the custom oligonucleotide data sheet from the primer manufacturer. The recommended annealing temperature from the manufacturer is around 52 degrees Celsius (specifically 51.8 and 53.8 degrees Celsius for different sequences). As a result, we chose 52 degrees Celsius as our annealing temperature according to MIQE standards.  The information was added to our manuscript on lines 457-459

Comment 7: Does the presentation of qPCR data comply with MIQE standards?

Response 7: All real-time PCR experiments were duplicated from at least three independent treatments. The relative gene expression was calculated as previously described and by the MIQE guidelines. The amplification efficiencies of qPCR ranged from 95% to 105% for all the tested primer sets. The information was added to our manuscript on lines 461-463 with reference on lines 689-691

Comment 8: There are various treatment methods when conducting in vitro experiments
with crude extracts of natural medicines. Direct application of crude extracts to cell lines may not mimic the physiological processes of drugs in vivo. Why didn't the authors choose to use serum containing the drug for treating cell lines?

Response 8: We appreciate the previous opinions from the reviewers. The components in serum include proteins, amino acids, sugars, vitamins, and peptides. The crude extract cannot be completely dissolved, resulting in uneven concentration. Therefore, we used DMSO solvent to dissolve it. Additionally, we aimed to prevent interference from the components in the serum in our results from the treatment of Salvia miltiorrhiza. Consequently, we opted to use crude extracts instead of serum containing the drug for treating the cell line. The information was added to our manuscript on lines 282-286

Reviewer 3 Report

Comments and Suggestions for Authors

The paper shows the effects of Salvia miltiorrhiza Bunge  crude extract in the prevention of fibrosis and inflammation during diabetic nephropathy, both in vitro and in vivo. The paper is well written but needs some improvements:

1)the HPLC profile does not show the percentages of individual compounds. it is essential to understand better the effect.

2) in the title it is well highlighted that S. miltiorrhiza  modulates the inflammatory effect, but it would be necessary to evaluate other cytokines, only one appears unconvincing for this statement

3) Please check the resolution of the figure 

Discussion:

Line 252: add some references about outher treatment (ex  doi: 10.3390/jcm9051600.)

 line 260: you suggest protective effects against oxidative stress, but you have not evaluated oxidative stress, why? It would be nice to add this data

Line364-365: delete line

Line 427. Please, add more information about the primary antibody

Line 449: add reference of the dose used in mouse

Author Response

Author’s reply to Reviewer 3

The paper shows the effects of Salvia miltiorrhiza Bunge crude extract in the prevention of fibrosis and inflammation during diabetic nephropathy, both in vitro and in vivo. The paper is well written but needs some improvements:

Comment 1: the HPLC profile does not show the percentages of individual compounds. it is essential to understand better the effect.

Response 1: Thank you for your precious advice. We acknowledge that the HPLC profile cannot provide the precise percentage; however, it gives us a hint about the main component in the layer we are interested in. In the future, we will conduct research to analyze the important pure compounds provided by the HPLC profile of the EtOAc layer to clarify which compound is capable of diminishing fibrosis in diabetic nephropathy. We added on information of the passage number on lines: 260-263

Comment 2: in the title it is well highlighted that S. miltiorrhiza  modulates the inflammatory effect, but it would be necessary to evaluate other cytokines, only one appears unconvincing for this statement,

Response 2: We apologize for the inappropriate word usage in our title. Since we did not provide the cytokine data for the anti-inflammatory effect of S. miltiorrhiza, we will delete the word ''inflammatory'' in our title to better align with the article content.

Comment 3: Please check the resolution of the figure 

Response 3: Thank you for your valuable feedback. We will improve the image resolution.

Comment 4: Line 252: add some references about other treatment (ex  doi: 10.3390/jcm9051600.)

Response 4: Thanks for your previous suggestions. We will provide some valuable references about the current treatments for DM nephropathy and the possible challenges of current treatment strategies. We added reference information on lines: 653-661

Comment 5: Line 260: you suggest protective effects against oxidative stress, but you have not evaluated oxidative stress, why? It would be nice to add this data

Response 5: From our available data, indeed, we cannot tell our effect can against oxidative stress, we acknowledge that the description is incorrect. We will remove the content of protective effects against oxidative stress (line: 290)

Comment 6: Line364-365: delete line

Response 6: We sincerely apologize for not thoroughly reviewing the content, which resulted in this typographical error. We have corrected the manuscript as your advice.

Comment 7: Line 427. Please, add more information about the primary antibody

Response 7: We apologize for the incomplete information regarding the antibodies used in our study. The relevant information including catalog numbers, and dilution factors will be added to the manuscript on lines: 471-476

Comment 8: Line 449: add reference of the dose used in mouse

Response 8: Thanks for the reviewer's previous feedback. We have researched the relevant study on STZ-induced DM mice with ethyl acetate extract of Salvia miltiorrhiza (DOI: 10.1142/S0192415X17500781). However, the current study lacks information on the dose for intraperitoneal injection. Another study mentioned that intraperitoneal injection of Salvia miltiorrhiza extract is not suitable for our topic (DOI: 10.3390/antiox9090857). We conducted a dose-dependent trial, testing 5/10/15 mg/kg doses of EtOAc extract, and found that the 10mg/kg dose was the minimal dose that yielded the maximum effect. Therefore, following the trial, we selected 10mg/kg as our intraperitoneal injection dose. We added on information of the passage number on lines: 490-491

Round 2

Reviewer 2 Report

Comments and Suggestions for Authors

The author has addressed my concerns very well. I appreciate all the efforts put in by the authors.